# COVID-19-Induced Acute Respiratory Distress Syndrome Treated with Hyperbaric Oxygen: Interim Safety Report from a Randomized Clinical Trial (COVID-19-HBO)

**DOI:** 10.3390/jcm12144850

**Published:** 2023-07-24

**Authors:** Anders Kjellberg, Johan Douglas, Adrian Hassler, Sarah Al-Ezerjawi, Emil Boström, Lina Abdel-Halim, Lovisa Liwenborg, Eric Hetting, Anna Dora Jonasdottir Njåstad, Jan Kowalski, Sergiu-Bogdan Catrina, Kenny A. Rodriguez-Wallberg, Peter Lindholm

**Affiliations:** 1Department of Physiology and Pharmacology, Karolinska Institutet, 171 77 Stockholm, Sweden; 2Perioperative Medicine and Intensive Care Medicine, Karolinska University Hospital, 171 76 Stockholm, Sweden; 3Department of Anaesthesia and Intensive Care, Blekingesjukhuset, 371 85 Karlskrona, Sweden; 4Acute and Reparative Medicine, Karolinska University Hospital, 171 76 Stockholm, Sweden; 5JK Biostatistics AB, 113 35 Stockholm, Sweden; 6Department of Molecular Medicine and Surgery, Karolinska Institutet, 171 76 Stockholm, Sweden; 7Academic Specialist Center, Center for Diabetes, 113 65 Stockholm, Sweden; 8Department of Oncology and Pathology, Karolinska Institutet, 171 64 Stockholm, Sweden; 9Department of Reproductive Medicine, Division of Gynaecology and Reproduction, Karolinska University Hospital, 171 76 Stockholm, Sweden; 10Department of Emergency Medicine, Division of Hyperbaric Medicine, University of California San Diego, La Jolla, CA 92093, USA

**Keywords:** COVID-19, respiratory distress syndrome, ARDS, hyperbaric oxygen therapy, HBOT, oxygen toxicity, safety, clinical trial

## Abstract

Background: A few prospective trials and case series have suggested that hyperbaric oxygen therapy (HBOT) may be efficacious for the treatment of severe COVID-19, but safety is a concern for critically ill patients. We present an interim analysis of the safety of HBOT via a randomized controlled trial (COVID-19-HBO). Methods: A randomized controlled, open-label, clinical trial was conducted in compliance with good clinical practice to explore the safety and efficacy of HBOT for severe COVID-19 in critically ill patients with moderate acute respiratory distress syndrome (ARDS). Between 3 June 2020, and 17 May 2021, 31 patients with severe COVID-19 and moderate-to-severe ARDS, a ratio of arterial oxygen partial pressure to fractional inspired oxygen (PaO_2_/FiO_2_) < 26.7 kPa (200 mmHg), and at least two defined risk factors for intensive care unit (ICU) admission and/or mortality were enrolled in the trial and randomized 1:1 to best practice, or HBOT in addition to best practice. The subjects allocated to HBOT received a maximum of five treatments at 2.4 atmospheres absolute (ATA) for 80 min over seven days. The subjects were followed up for 30 days. The safety endpoints were analyzed. Results: Adverse events (AEs) were common. Hypoxia was the most common adverse event reported. There was no statistically significant difference between the groups. Numerically, serious adverse events (SAEs) and barotrauma were more frequent in the control group, and the differences between groups were in favor of the HBOT in PaO_2_/FiO_2_ (PFI) and the national early warning score (NEWS); statistically, however, the differences were not significant at day 7, and no difference was observed for the total oxygen burden and cumulative pulmonary oxygen toxicity dose (CPTD). Conclusion: HBOT appears to be safe as an intervention for critically ill patients with moderate-to-severe ARDS induced by COVID-19. Clinical trial registration: NCT04327505 (31 March 2020) and EudraCT 2020-001349-37 (24 April 2020).

## 1. Introduction

Severe COVID-19 often presents as an inflammatory condition in the lungs (resembling organizing pneumonia) with vascular endothelitis. The original aim of this study was to use the anti-inflammatory effects of hyperbaric oxygen therapy (HBOT) to prevent intubation and save intensive care unit (ICU) beds. In a wider perspective, we investigated the use of hyperbaric oxygen (HBO_2_) in cases of severe pneumonia near respiratory failure. Despite the decreased mortality as regards severe COVID-19, there is a need for additional safe and effective treatments for patients developing acute respiratory distress syndrome (ARDS) [1], and the results may be useful in applications beyond COVID-19.

HBOT consists of breathing 100% oxygen above the normal atmospheric pressure, raising the inspired partial pressure of oxygen (PO_2_) beyond 101.3 kPa to as high as 280 kPa. This process greatly increases oxygen transfer and delivery through diffusion barriers. In addition to improved gas delivery, the high PO_2_ has specific biological effects, i.e., it reduces inflammatory cytokines via several transcriptional factors, including hypoxia-inducible factor 1 (HIF-1) [2,3]. By attenuating the nuclear factor kappa-light-chain-enhancer of activated B cells (NFkB), possibly through HIF-1, HBOT has the ability to restore inflammatory homeostasis [4]. HBOT is used in clinical practice for several inflammatory conditions, such as radiation injury [5,6,7], flares of ulcerative colitis [8], and diabetic foot ulcers [9]. HBOT is associated with reduced mortality when used as an adjuvant therapy in severe bacterial infections, including necrotizing soft-tissue infections [10] and brain abscesses [11].

Although HBOT has virtually no relevant side effects in the regular patient population, patients with severe COVID-19 are often supported by high-flow oxygen 24/7, creating a risk of pulmonary oxygen toxicity (POT) that could, potentially, be accelerated by HBOT [12]. Severe COVID-19 patients differ from other ARDS patients by often presenting with ‘happy hypoxemia’ (hypoxemia in the absence of dyspnea, suggesting an adaption to low PO_2_). Hence, patients might be at greater risk of POT if liberal goals for the arterial partial pressure of oxygen (PaO_2_) (11–13 kPa) are targeted, and possibly with even more conservative goals of 8–10 kPa [13]. Another concern is potential barotrauma from gas expansion on decompression. Healthy lungs have traditionally been a requirement for diving and unpressurized flight. In clinical practice, patients with severe chronic obstructive pulmonary disease, pulmonary fibrosis, or cystic disease are normally excluded from HBOT [14]. There is an ongoing debate regarding whether clinical equipoise exists for HBOT in ICU patients with severe pulmonary disease [15].

The use of HBOT for severe COVID-19 was first demonstrated in a case study from Wuhan, China [16]. Additional reports, including two randomized clinical trials, were published during the pandemic that supported the potential positive effects of HBOT while demonstrating no increase in adverse events (AEs) with HBOT [17,18,19,20,21,22]. Several hypotheses with the common denominator of an ‘anti-inflammatory effect’ have been postulated [23,24].

In this article, we report on the safety profile of HBOT to guide other researchers in trial design and to support clinicians who may consider HBOT for compassionate use in critical-care patients with pathological lung tissue.

## 2. Materials and Methods

### 2.1. Study Design

This phase II multicenter, randomized, controlled, parallel-arms, open-label clinical trial to evaluate the safety and efficacy of HBOT for severe COVID-19 was conducted at three sites: Blekingesjukhuset, Karlskrona, and Karolinska University Hospital, Stockholm, both in Sweden, and St Caritas University Hospital Regensburg, in Germany. The trial was investigator-initiated and sponsored by the Karolinska Institutet, Solna, Sweden. Follow-up was carried out for 30 days from randomization. The trial protocol was published previously [25].

### 2.2. Participants

The patients were recruited directly on hospital wards. The inclusion criteria were as follows: aged 18–90 years, moderate-to-severe ARDS induced by COVID-19, a ratio of arterial oxygen partial pressure to fractional inspired oxygen (PaO_2_/FiO_2_) of <26.7 kPa (200 mmHg), and at least two defined risk factors for ICU admission and/or mortality. Patients with severe COPD, significant pulmonary fibrosis, or other contraindications for HBOT were excluded [25]. 

### 2.3. Randomization

The subjects were enrolled and randomized consecutively as they were found to be eligible for inclusion in the study. Randomization was performed in a 1:1 allocation, stratified by site and gender in blocks to either HBOT + best practice, or best practice. The randomization sequence was computer-generated using an internet-based application RANDOMIZE.NET, which ensured that the outcome of the randomization, i.e., treatment group, were masked until the time point at which each subject was to be randomized.

### 2.4. Procedures

All patients in the HBOT group received treatments consisting of 60 min at 2.4 atmospheres absolute (ATA) with 10 min compression/decompression time and one air-break, making the total treatment time approximately 80 min.

The full procedure list was published previously and is available online [25]. Within 24 h of randomization, the subjects allocated to HBOT received their first treatment. The subjects then received a maximum of five treatments within seven days of randomization. The national early warning score (NEWS) and PaO_2_/FiO_2_ (PFI) were recorded three times a day for both groups, while the HBOT group had NEWS/PFI recorded before/after the treatment. Both groups were managed according to best practice including nasal high-flow oxygen, non-invasive ventilation, and awake prone positioning made at the discretion of the treating physician. The subjects received medical treatment including corticosteroids and low-molecular-weight heparin (LMWH). All concomitant medications including normobaric oxygen were recorded. AE were recorded and evaluated according to protocol as AE or serious adverse events (SAE) and graded as mild, moderate, or severe. Causality in relation to HBO_2_ was also assessed. Staff safety was evaluated using reports made via the hospitals’ reporting system for reporting negative events. The mean oxygen dose was recorded three times daily over an eight-hour period and the cumulative ’oxygen burden’ was calculated on a daily basis and longitudinally over the course of the trial (30 days). Oxygen is considered toxic for healthy subjects at any fraction of inspired oxygen (FiO_2_) > 0.5 ≅ PO_2_ > 50 kPa [26]. Oxygen toxicity is traditionally calculated as units of pulmonary toxic dose (UPTD), with repeated exposures expressed as cumulative pulmonary toxic dose (CPTD). We used a simplified calculation derived from the traditional equation: UPTD = t × [0.5/(PO_2_ – 0.5)]^−5/6^, with PO_2_ in ATA and time in minutes [27]. The simplified equation for clinical use is defined as UPTD_ICU_ = t × (FiO_2_ − 0.5), and summarized daily UPTD values were used to calculate the total ‘dose’ of oxygen above FiO_2_ 0.5 (CPTD_ICU_). In the HBOT group, one hour of mean oxygen was replaced with the ‘HBOT dose’.

To reduce bias, the data were monitored by an independent monitor that checked the source data for all AE and selected the source data for exploratory endpoints according to the monitoring plan.

### 2.5. Outcomes

The results of the primary endpoints and main secondary endpoints have not yet been analyzed. The trial protocol, including safety endpoints, has been described previously and is available online [25]. The safety endpoints included the following: the number of subjects, proportion of subjects and number of events of AE/SAE/serious adverse drug reaction (SADR); mean change in PFI before and after HBOT compared with mean variance in PFI in the control group on days one to seven; and mean change in NEWS before and after HBOT compared with mean change in daily NEWS in the control group on days one to seven. Any negative events experienced by the chamber staff associated with the treatment of subjects were also recorded.

The exploratory outcomes associated with safety analysed in the interim analysis included the following:Mean oxygen dose per day including HBO_2_ and cumulative pulmonary oxygen toxicity expressed as UPTD and CPTD from day 1 to day 30.Number of secondary infections, number of events and patients from day 1 to day 30.Diagnosed PE needing treatment, number of events and patients from day 1 to day 30.

### 2.6. Statistical Analysis

Safety analyses were performed on the safety population as shown in the consolidated standards of reporting trials (CONSORT) flow diagram, Figure 1.

The safety endpoints included the following: AE, vital parameters (NEWS), and oxygenation (PFI). Statistical analysis for the NEWS and PFI scores was performed using analysis of covariance (ANCOVA) including baseline levels as a covariate, and treatment as a fixed factor in the models. The null hypothesis was no difference between the treatment groups. Tests were two-sided with a type I error rate of 0.05, where *p* < 0.05 was regarded as statistically significant. There was no adjustment for multiplicity as the safety endpoints and corresponding results were regarded as exploratory. Analysis was performed on the safety population with the observed data.

A descriptive analysis of the number and percentage of patients reporting AE, and the number of AE reported are presented. SAE are also presented in separate tabulations. The events are tabulated according to system organ class and preferred term.

All continuous safety variables, such as age, body mass index (BMI), days with symptoms, and number of risk factors were described using summary statistics.

All categorical variables, such as ethnicity and smoking habits, were summarised using frequencies and percentages.

## 3. Results

Between 3 June 2020, and 17 May 2021, 31 patients were included in the study; of the 54 patients assessed for eligibility, 31 subjects were randomized, 15 to the HBOT group and 16 to the best practice group. One patient in the HBOT group was excluded from analysis due to withdrawal of consent before the first treatment. One patient was excluded from the control group due to a negative SARS-CoV-2 test and was positive for adenovirus. A CONSORT flow diagram for the safety analysis is shown in Figure 1. Three subjects died during the trial: two in the HBOT group and one in the control group. The primary outcome (ICU admission) has not yet been analyzed, with the cutoff date for inclusion in this interim safety analysis being 1 October 2021. The subjects had moderate-to-severe ARDS at inclusion and groups were balanced at baseline (Table 1).

Those who received HBOT had a greater numerical improvement in NEWS and PFI at days 7, 14, and 30. The changes were not statistically significant except for change in PFI at day 14 (*p* = 0.023); however, this was not a predefined safety endpoint (Figure 2).

A total of 95 AE were registered; of the 23 SAE, 9 (in six subjects) were in the HBOT group and 14 (in six subjects) in the control group. Hypoxia was the most commonly reported AE, with a slightly different distribution, and grade of AE. One SAE (hypoxia) coincided with HBO_2_ treatment and led to intubation within one hour after HBOT; as this event was possibly related to HBO_2_, it was assessed as a SADR even though the ICU admission was planned before the treatment. (Table 2). There were no negative events reported in the chamber staff treating the subjects (e.g., contact with subject aerosols). A complete list of adverse events with Medical Dictionary for Regulatory Activities (MedDRA) coding, including system organ class (SOC), the preferred term (PT) and code, is available as Appendix A.

An analysis of several predefined exploratory endpoints associated with safety was carried out as follows:Oxygen toxicity and total oxygen dose in a subgroup (all patients from the Karolinska University Hospital, n = 20): The cumulative oxygen burden expressed as CPTD_ICU_ was not significantly different between the groups with the mean (SD) HBOT 1618 (1791) and best practice 1724 (2374), (*p* = 0.882) (Figure 3A). There was a trend towards faster recovery in terms of days with supplemental oxygen in the HBOT group (*p* = 0.318) (Figure 3B).Secondary infections: two ventilatory-associated pneumonias (VAPs), two bacteraemia/sepsis and one abscess in m. obturatorius in the control group. One urinary tract infection in the HBOT group.Thrombotic events: One patient in the HBOT group had a small pulmonary embolism.Barotrauma: One subject in the control group had a pneumothorax. Five subjects had pneumomediastinum, four in the control group and one in the HBOT group.

Illness severity is illustrated using a typical example of a CT chest scan in Figure 4.

## 4. Discussion

This interim report focuses on safety endpoints and the exploratory endpoints related to safety while using HBOT to potentially treat COVID-19, with a focus on oxygen toxicity and barotrauma, considering these are of major concern. To the best of our knowledge, this is the first report on the safety of HBOT in patients with severe COVID-19 in compliance with ICH-GCP. There was a trend towards a lower NEWS and higher PFI values, with a statistically significant difference in PFI at day 14 (Figure 2). There was no statistically significant difference in the occurrence of AE/SAE or barotrauma between the groups, with numerically more AE/SAE and barotraumas in the control group. Despite the added exposure of HBOT in the first seven days, there was no difference in CPTD_ICU_, and there was a trend towards a lower probability of need for supplemental oxygen. Although the study was not designed to evaluate efficacy and the main efficacy endpoints have not been analyzed yet, the data show a trend towards a benefit to patients with HBOT, as based on the safety endpoint results together with the exploratory endpoints related to safety.

The SARS-CoV-2 pandemic has led to an unparalleled number of ARDS patients. Thus, in order to target COVID-19 patients at risk of mortality, a more pragmatic definition than the Berlin definition for COVID-19-induced ARDS has been suggested; this is outlined by a need for nasal high-flow oxygen (NHFO) of FiO_2_ > 0.35 and ≥20 L/min, with a 5–14 day window, and unilateral opacities [28]. There is an ongoing debate regarding whether COVID-19-induced acute respiratory failure (C-ARF) should be treated as a separate entity from traditional ARDS [29,30]. A recent Delphi expert consensus statement agreed that the pathophysiology of C-ARF is similar to that of ARDS [31].

Some major concerns associated with the use of HBOT in ARDS patients include the risk of barotrauma, absorption atelectasis, and POT, which might constitute arguments against the use of HBOT in severe COVID-19 [12]. As there was a fear of oxygen toxicity occurring during treatment when the trial was designed early in the pandemic, the treatment protocol had a wide range of parameters to be decided on at the discretion of the treating physician (1.6–2.4 ATA for 30–60 min with 5–10 min compression time and 5–10 min decompression time). All subjects were eventually treated with 2.4 ATA for 80 min.

Previously published RCTs reporting on the use of HBOT for severe COVID-19 have some major differences to our trial. Most importantly, the Argentinian study (Cannellotto et al.) used a ‘soft-shell’ Revitalair 430 chamber at a pressure of 1.45 ATA, utilising oxygen delivered via a non-rebreather mask [19]. This study does not report whether medical oxygen 100% or an oxygen concentrator was used; with only the use of the equipment described in the report, the patient could not receive 100% oxygen, and the expected range would be between 60 and 91% O_2_. If an oxygen concentrator was used, there also would have been an accumulation of argon of 2–4%. There was no information regarding PFI or ARDS either for inclusion in the study or in the outcome measures reported; thus, drawing any conclusions from this study regarding the safety of HBOT in patients with ARDS is not possible. A Polish trial (Siewiera et al.) used a similar protocol to ours, including the enrolment of patients with moderate ARDS, treatment with medical oxygen 100% at 2.5 ATA/80 min, and the measurement of NEWS and PFI; however, no AE were reported [21]. Previous RCTs were well-designed but not conducted in compliance with ICH-GCP; hence, the number of AE may have been underreported.

Most subjects in our trial were administered non-invasive mechanical ventilation (NIV) or nasal high-flow oxygen (NHFO), but we did not treat any patients with invasive mechanical ventilation with HBO_2_. Barotrauma during mechanical ventilation is well-known to all intensivists and there is a strong consensus regarding pressure and volume limits in ARDS [31]. There was no evidence of increased barotrauma in the HBOT group; in fact, pneumothorax and pneumomediastinum were more frequent in the control group. The barotraumas seen in the present trial were most likely caused by positive pressure ventilation and not related to HBOT. It has been previously suggested by a prospective observational study that HBO_2_ can be administered safely in mechanically ventilated patients with ARDS; however, this action has not been evaluated in a randomized controlled trial [32].

Ventilation with FiO_2_ 1.0 is suggested to cause absorption atelectasis [33]. It has been debated whether absorption atelectasis is clinically relevant in critically ill patients with an existing raised FiO_2_ [34]. In the present study, there was no difference in the number of AE related to hypoxia between the groups. Only one of these occurred within six hours of HBOT; hence, it was evaluated as being possibly related to the HBO_2_ treatment and reported as a SADR. Our data do not suggest that there was an increased risk of atelectasis post HBO_2_ treatment. We suggest that the worry of desaturation due to absorption atelectasis is greatly overrated, as there was only one SAE (hypoxia leading up to intubation) that could be related to HBOT. This specific subject was on NIV with a FiO_2_ = 0.8 before HBOT and the ICU admission was planned before the treatment. To evaluate oxygen toxicity, we recorded the mean daily dose of oxygen, which is rarely carried out in clinical practice.

The toxic effects of oxygen upon the lungs were discovered more than a century ago [35]. POT can be divided into two phases, an early exudative and a late proliferative phase [36]. These two phases have been experienced clinically by most intensivists treating ARDS patients, where the early phase is reversible and the latter, which leads to fibrosis, is irreversible if the oxygen fraction cannot be lowered below a toxic dose [37]. Even though POT is well-known, some of the damage seen in the ICU may be iatrogenic. Supplemental oxygen has also been associated with negative effects on many other organs, which have implications for the critically ill patients [38]. Most intensivists would avoid FiO_2_ = 1.0 [31] but the POT limit of FiO_2_ > 0.5 accepted in diving and hyperbaric medicine is rarely discussed in intensive care medicine; rather, oxygen toxicity is targeted at PaO_2_ or peripheral saturation (SpO_2_) [13]. Efforts have been made to establish a dose–response equation for the toxic effects, but there is not a linear correlation between the two [39]. The equation is extrapolated from mice, tested on healthy individuals, and there are numerous other factors apart from the concentration × time integral [39,40]. Despite attempts to measure or better predict POT, the gold standard for healthy divers is still calculated using the traditional equation (see the Methods section) [27]. For clinical use in the critical care setting, we suggest the use of the following simplified equation: UPTD_ICU_ = t × (FiO_2_ − 0.5). This equation can be easily calculated in order to obtain an estimate of POT. It is well-known that intermittent reductions in oxygen, or ‘air-breaks’, reduce harm and may even be beneficial [41,42]. Daily UPTD_ICU_ values can be collected easily and summarised as CPTD_ICU_ to evaluate the risk of pulmonary toxicity over time.

Even though UPTD_ICU_ is not an exact calculation, it provides a rough estimate that can be used when evaluating POT in clinical trials. We further calculated the cumulative oxygen burden, as it is unknown whether any FiO_2_ above normal air (21% O_2_) is toxic to injured lungs. If UPTD_ICU_ is calculated daily, we speculate that it may also be beneficial to use HBO_2_ in patients on invasive mechanical ventilation to reduce inflammation and to reduce the cumulative oxygen burden.

The present trial has several limitations. It was an open-label trial; therefore, neither the patients nor the investigators were blinded to the allocated treatment. Both groups were treated according to best practice by independent staff, with the addition of a maximum of five HBO_2_ treatments for subjects allocated to HBOT + best practice. Due to logistical reasons, it was not possible to conduct a single-blinded trial and as the safety of HBO_2_ for this indication has not been evaluated previously, we chose an open-label design. In this multicentre trial, the subjects were enrolled in different phases of the pandemic. The first eight subjects were enrolled before corticosteroids were considered best practice and this may have affected the outcome of these subjects negatively. However, the use of corticosteroids to treat ARDS is still being debated due to multiple negative effects, such as hyperglycaemia, infection, and weakness [43]. A similar incidence of SAE was found in this cohort of subjects compared to those enrolled later in the trial and they were equally distributed between the two groups; interestingly, however, secondary infections were more frequent in the control group, the subjects of which had all received concomitant corticosteroids. Most subjects were included during the second and third wave, though none of the subjects were fully vaccinated, a factor that may have affected the outcome of this analysis and may affect later efficacy analyses when subjects from subsequent waves are included. Due to the small sample size of this interim analysis, the results will also be prone to both type I and II errors, despite the fact that the group demographics appear similar, and the trial is randomized. There may be other confounding factors not accounted for.

## 5. Conclusions

Based on the interim safety analysis of our randomized controlled trial, we propose that HBOT is well-tolerated and can be safely used as an intervention for critically ill patients with moderate-to-severe ARDS induced by COVID-19. We speculate that HBOT may be useful in ARDS caused by conditions other than COVID-19; for example, it may be used in mechanically ventilated patients to reduce the cumulative oxygen burden. Larger randomized controlled trials are warranted to confirm the safety and evaluate the efficacy of this treatment.

## Figures and Tables

**Figure 1 jcm-12-04850-f001:**
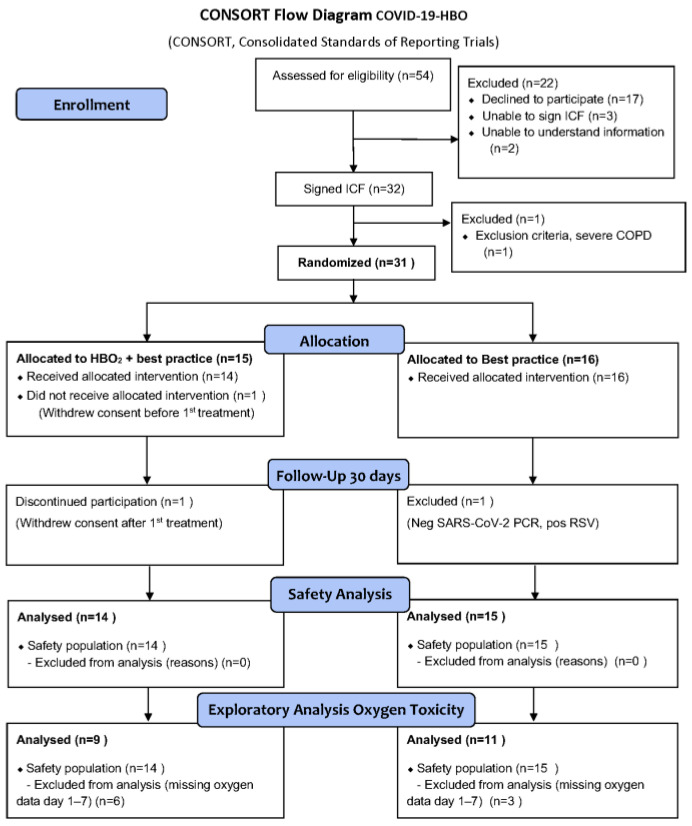
CONSORT flow diagram for safety analysis.

**Figure 2 jcm-12-04850-f002:**
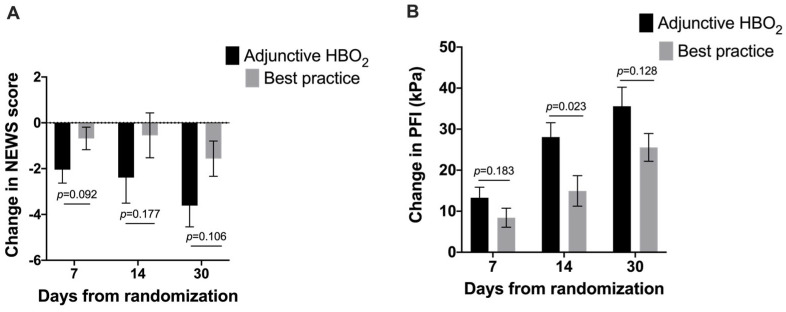
Changes from baseline in NEWS (**A**) and PFI (**B**) day 7, day 14 and day 30 (Mean and SD).

**Figure 3 jcm-12-04850-f003:**
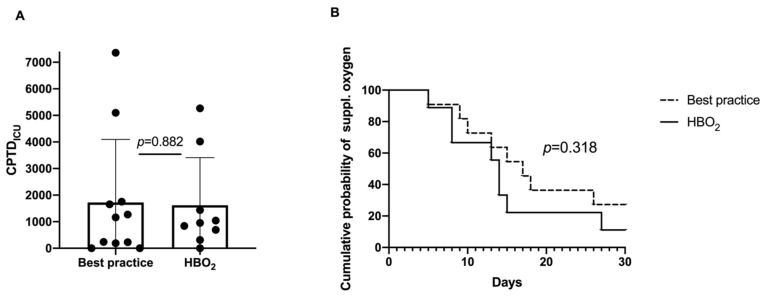
Cumulative oxygen burden expressed as CPTD_ICU_ day 1 to day 30 (mean [SD]) (**A**), and Kaplan–Meier curve describing the cumulative probability of need for supplemental oxygen day 1 to day 30 (**B**).

**Figure 4 jcm-12-04850-f004:**
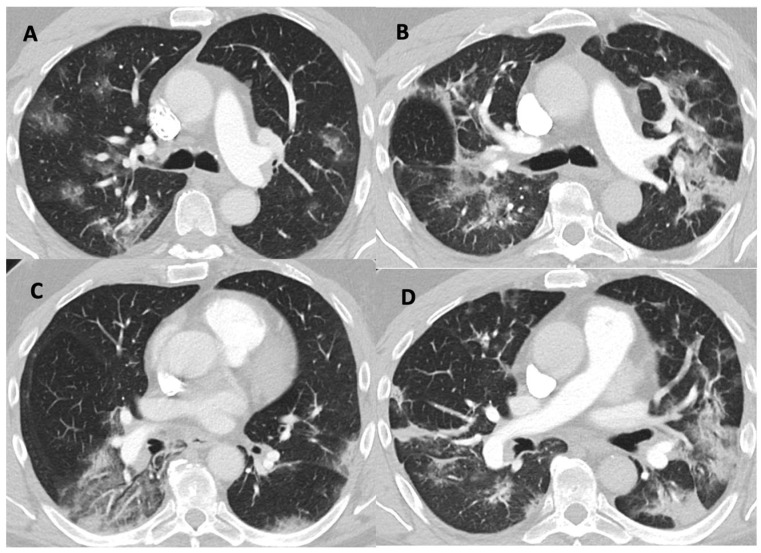
Typical parenchymal changes for COVID-19 with progression showing that HBO treatments were given to patients with advanced parenchymal disease. Computed tomography pulmonary angiography. Two locations showed with 3.1 mm/ WL −400 WW 1600. Left panels, four days prior to first HBOT (**A**,**C**). Right panels, corresponding locations two days after the fifth HBOT (**B**,**D**).

**Table 1 jcm-12-04850-t001:** Baseline characteristics expressed as mean (SD), or number (%) for the safety population.

Baseline Variable	HBOT + Best Practice N = 14	Best Practice N = 15
Age	67.4 (10.8)	63.3 (8.2)
Male sex	8 (57.1%)	8 (53.3%)
Caucasian ethnicity	13 (92.8%)	15 (100%)
BMI	29.4 (4.5)	29.2 (5.0)
Number of risk factors	2.93 (0.96)	3.13 (1.06)
Smoker (every day)	1 (7.1%)	0 (0%)
Former smoker	5 (35.7%)	5 (33.3%)
Never smoked	8 (57.1%)	10 (66.7%)
Time since initial symptoms (days)	9.93 (3.58)	11.67 (3.62)
NEWS at randomization	5.3 (2.0)	5.4 (1.7)
PFI at randomization	14.0 (3.5)	17.3 (6.4)

**Table 2 jcm-12-04850-t002:** AE overview.

Group	HBOT (N = 14) n (%) AE	Best Practice (N = 15) n (%) AE
Adverse events	14 (100%) 40	13 (87%) 55
Serious adverse events	6 (43%) 9	6 (40%) 14
Severe adverse events	3 (21%) 4	2 (13%) 3
Deaths	2 (14%)	1 (7%)
Life-threatening	1	2
Persistent or significant disability/incapacity	2	0
Initial or prolonged hospitalization	5 (36%) 7	5 (33%) 9
Congenital anomaly/birth defect	0	0
Relationship to IMP *—possible	2	0
Action taken regarding IMP (discontinued)	2	0

* Investigational medical product

## Data Availability

The full study protocol, statistical plan and consent form will be publicly available. The data will be available at a patient level; the data will be pseudonymized, and the full dataset and statistical code will be available upon request. A full description of the intended use of the data must be sent to the corresponding author for review and approval. Participant consent for data sharing is conditioned and new ethics approval may be required.

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
