# Peer review of "COVID-19-Induced Acute Respiratory Distress Syndrome Treated with Hyperbaric Oxygen: Interim Safety Report from a Randomized Clinical Trial (COVID-19-HBO)"

_jcm, 2023, doi:10.3390/jcm12144850_

Round 1
Reviewer 1 Report
The manuscript says: ”The original aim of this study was to use the anti-inflammatory effects of Hyperbaric Oxygen Therapy (HBO2) to prevent intubation and save ICU-beds” and further:
” The results of the primary and main secondary endpoints have not yet been analysed. ”
They have probably found that additional (secondary) endpoints, including safety of the treatment might be worth of publishing.
However, every physician working with hyperbaric oxygenation or pressure medicine in general is well aware of the existence of the risk of pulmonary toxicity (from 24/7 O2 delivery) and also know that in routine HBO2 practice it can be prevented by appropriate air breaks. The same applies to pulmonary and other barotraumas.
Therefore, overall, the manuscript, as it is now, contains no essential message, and the main (safety) issues that have been covered are of minor interest.
In addition, its main message seems to strengthen the unreasonably prevalent ”oxygenphobia” or ”HBO2phobia” particularly by the anesthesiologists.
However, the manuscript could be useful if the authors discuss the following issues:
1.
HBO2 can correct (tissue) hypoxia
The authors write: ”In addition to frank gas delivery the high PO2 has specific biological effects; it reduces inflammatory ….”
From this one could conclude that the main point of treating with HBO2 is to reduce inflammation, not to correct hypoxia- particularly in the brain. In general, the authors do not seem to connect hypoxia and inflammation. A connection known already for almost 50 years (Fridovich, I. Oxygen is Toxic! Bioscience 1977, 27, 462-466; Fridovich, I. Hypoxia and oxygen toxicity. Adv. Neurol. 1979, 26, 255-259). One reason is prbably that the authors seem to see oxygen primarily as a ”drug” (”Hyperbaric oxygen (HBO2) is a possibly effective drug..”)
(Is this because the ms comes from the Dept of Pharmacology ??).
2.
The authors mention other studies: ” Additional reports including two randomised clinical trials have been published during the pandemic supporting potential positive effects while not demonstrating any increase in adverse events (AE) for HBOT[17-22]”.
The outcomes/results of at least some of those- with identical aims- should be compared and discussed. E.g., Thibodeaux, K., et al., (J Wound Care, 2020) described only 5 cases with ”dramatic improvement”. Cannellotto, M., et al., (Emerg Med J, 2021) were able to recruit 40 patients before the study was interrupted after an interim analysis that revealed also a dramatically more rapid correction of hypoxaemia/hypoxia in HBO2 group compared to controls.
This ms has 30 cases – 6 x more than Thibodeaux- ”with a trend to improvement”. Is this, probably less favourable outcome compared above earlier studies, due to time delay (about 9 days) from the start of the disease or are there other assumed causes, should be discussed.
3.
One additional crucial question that should be discussed : If there is an acute illness of which the main end-point is death, and the treatment is oxygen (which certainly is not a drug but an elixir of life and without which any healing is not possible), how ethically acceptable it is to select patients for the placebo group (and not give oxygen as efficiently as possible for severely hypoxic patients).
4.
The authors seems to have the opinion that COVID-19 is purely a lung disease- and that patients with COVID-19 hypoxemia have ARDS, already from the first days of the disease. This is not true, and patients with ”happy hypoxia” most probably do not have ARDS (yet). Both early neurological symptoms and Long Covid clearly show that the virus hits brain at an early stage.
Reviewer 2 Report
The manuscript by Anders Kjellberg et al. reports the safety and efficacy of HBOT for severe COVID-19 in critically ill patients with moderate acute respiratory distress syndrome (ARDS). The interim data from a randomised controlled, open label, clinical trial registration: NCT04327505 (March 31, 2020) and EudraCT 2020-001349-37 (April 24, 2020) is described, the protocol for which has been previously published (BMJ Open, 2021. 11(7): p. e046738).
The safety profile for HBOT used to treat conditions of the lung are of particular interest as pulmonary conditions often mean patients are excluded from receiving HBOT on safety grounds. The debate is ongoing and more research data is certainly needed.
The manuscript is well written and clear, with only a few suggestions detailed below. The conclusions of the manuscript are sound and Consort guidelines adhered to. The subject matter is certainly of interest and adds to current knowledge.
Suggestion to consider for improvement:
Abbreviations – abbreviations are prevalent throughout the manuscript but would benefit from review. The full description on first use is sometimes extended to the second, third and more uses eg Adverse Event (AE) is written in full and abbreviated on lines 33, 79, 118, 155 and 192. Others including COVID-19, PFI, NEWS are not written in full on first use. The accentuation of abbreviations by underlining the letter components (HBOT - L47) is probably not necessary.
Barotrauma – the potential for barotrauma from gas expansion during decompression from HBOT is described in the introduction (L70-71). Barotrauma was the 4th exploratory endpoint associated with safety (L203) and there were multiple instances of barotrauma. However, most cases were seen in the control group and therefore not caused by decompression but more likely caused by positive ventilation. In a manuscript involving hyperbaric oxygen therapy the authors may want to consider clarifying the source of barotrauma (if possible) to differentiate between hyperbaric pressure changes v. mechanical ventilation? This is touched upon in the discussion (L261) .
L69-70: ‘goals for arterial partial pressure of oxygen are targeted.’
L104: ‘secured’ – consider change to ‘ensured’
Table 1: Smoking status – the numbers for the HBO group total 15 (9+5+1) but n=16 ?
Table 1: Time since initial symptoms – what are the units (days?)?
L317: suggest new sentence ‘…… control group. They all ……’
L323: suggest new sentence ‘ trial is randomised. There may be ……’
Language generally very good.
Round 2
Reviewer 1 Report
The decisions/recommendations by UHMS, EUBS and ECHM can sometimes be rather far from the best interest of patients.
FDA definition (section 201g) of a drug: ..“articles (other than food) intended to affect the structure or any function of the body of man or other animals.”
See "other than food"